# HOXA9 Overexpression Contributes to Stem Cell Overpopulation That Drives Development and Growth of Colorectal Cancer

**DOI:** 10.3390/ijms23126799

**Published:** 2022-06-18

**Authors:** Brian Osmond, Caroline O. B. Facey, Chi Zhang, Bruce M. Boman

**Affiliations:** 1Cawley Center for Translational Cancer Research, Helen F. Graham Cancer Center & Research Institute, Newark, DE 19713, USA; bosmond@udel.edu (B.O.); caroline.facey@christianacare.org (C.O.B.F.); johnitch@udel.edu (C.Z.); 2Department of Biological Sciences, University of Delaware, Newark, DE 19713, USA; 3Department of Pharmacology & Experimental Therapeutics, Thomas Jefferson University, Philadelphia, PA 19107, USA

**Keywords:** HOX genes, cancer stem cells, colon cancer, retinoic acid signaling

## Abstract

HOX proteins are transcription factors that regulate stem cell (SC) function, but their role in the SC origin of cancer is under-studied. Aberrant expression of HOX genes occurs in many cancer types. Our goal is to ascertain how retinoic acid (RA) signaling and the regulation of *HOXA9* expression might play a role in the SC origin of human colorectal cancer (CRC). Previously, we reported that aldehyde dehydrogenase (ALDH) and other RA pathway components are co-expressed in colonic cancer SCs (CSCs) and that overpopulation of ALDH-positive CSCs occurs during colon tumorigenesis. Our hypothesis is RA signaling regulates HOXA9 expression, and dysregulated RA signaling results in HOXA9 overexpression, which contributes to CSC overpopulation in CRC. Immunostaining showed that HOXA9 was selectively expressed in ALDH-positive SCs, and HOXA9 expression was increased in CRCs compared to normal epithelium. Modulating RA signaling in CRC cells (HT29 and SW480) with ATRA and DEAB decreased cell proliferation and reduced HOXA9 expression. Bioinformatics analyses identified a network of proteins that functionally interact with HOXA9, and the genes that encode these proteins, as well as *HOXA9*, contain RA receptor binding sites. These findings indicate that the expression of HOXA9 and its functional network is regulated by RA signaling in normal colonic SCs, and, when dysregulated, HOXA9 may contribute to CSC overpopulation that drives CRC development and growth. Our study provides a regulatory mechanism that might be useful in developing treatments against CSC overpopulation in CRC.

## 1. Introduction

Cancer develops due to the dysregulation of diverse cellular processes. One potential source of this dysregulation is aberrant HOX gene expression, which is commonly found in several different cancer types [1,2,3,4]. HOX genes encode a group of highly conserved transcription factors that regulate embryonic development in humans and other animals. They are categorized into four highly conserved gene clusters named HOXA, HOXB, HOXC, and HOXD. These clusters are located on chromosomes 7, 17, 12, and 2, respectively. Outside of embryonic development, HOX genes also control the differentiation of adult stem cells (SCs). Because of their roles in embryonic development and stem cell differentiation, the dysregulation of HOX gene expression can disrupt normal SC function and promote tumorigenesis [1,2,3,4]. Furthermore, the ability of HOX proteins to promote abnormal SC function may, in turn, contribute to therapeutic resistance of cancer stem cells (CSCs) in several cancer types. Interestingly, a recent study identified a potential anticancer drug, 5H-pyro[3,2-a] phenoxazin-5-one (PPH), that targets the HOXC gene loci and the associated lncRNA *HOTAIR*, which are highly implicated in breast cancer [5]. HOXA9 overexpression in both colorectal cancer and leukemia is another example of the effects of HOX gene dysregulation [4,6,7,8] and is the focus of our current study.

The effects of HOXA9 overexpression are well documented in acute myeloid leukemia (AML), where said overexpression is linked to poor prognosis [9,10]. HOXA9 acts as a transcription factor for *NUP98* and other oncogenes. The overexpression of HOXA9 results in increased expression of these oncogenes, which in turn results in leukemia [10,11]. Several chemical agents that disrupt HOXA9 function, such as HXR9 peptide inhibitor and heterocyclic diamidine DNA ligands, are being used as potential leukemia treatments [10,11]. In addition to leukemia, HOXA9 overexpression is also observed in cases of colorectal cancer (CRC) [1,6].

While not as rigorously studied in adult solid tumors, *HOXA9* is linked to CRC, with overexpression of the *HOXA9* gene being associated with poor patient prognosis [12]. Indeed, overexpression of nucleus accumbens-associated protein 1 (NAC1) increased chemotherapy drug resistance to CRC cancer cells via induced HOXA9 overexpression [13]. HOXA9 is also found to be overexpressed in colon adenomas and alters the oncogenic mRNA profile [14]. However, this increased expression also inhibits cellular migration, indicating that HOXA9 is mainly involved in the adenomatous stage [14]. In fact, *HOXA9* gene knockdown shows no effect on CRC cell growth [12].

We have been studying whether HOX genes play a role in the SC origin of CRC. We discovered that *HOXA4*, *HOXA9*, and *HOXD10* have a role in regulating colonic SC renewal and are aberrantly expressed in cancer SCs (CSCs) [6,8,15]. In addition, other studies have reported that *HOXA9* inhibition by miR-133b suppresses metastasis [16]. Other HOX genes, particularly *HOXC8* and *HOXC9*, are also involved in colonic tissue SCs. Specifically, *HOXC8* and *HOXC9* promote the differentiation of CSCs [4]. *HOXA9* is also aberrantly expressed in other cancers [4,17]. For example, *HOXA9* is commonly expressed in hematopoietic stem cells and is downregulated when SCs differentiate [18]. In breast cancer, *HOXA9* overexpression activates signaling along the WNT pathway and is associated with a poor prognosis [17].

Retinoic acid (RA) is a derivative of vitamin A and a key component of the retinoid signaling pathway. In the RA pathway, all-trans retinoic acid (ATRA) is an RA pathway component produced when ALDH oxidizes retinol. ATRA production is catalyzed by the enzyme aldehyde dehydrogenase (ALDH) [19]. Administering exogenous ATRA drives RA signaling, while diethylamino benzaldehyde (DEAB), a known inhibitor of ALDH, inhibits downstream RA signaling. During activation of RA signaling, RA binds to a protein complex consisting of a retinoic acid receptor (RAR) and a retinoid X receptor (RXR). This complex then interacts with RAR–RXR binding sites found in several genes, including certain HOX genes [20,21]. The RA signaling pathway has been shown to have a key role in regulating SC function [22,23]. Colonic epithelial cells treated with ATRA have reduced proliferation and differentiate into neuroendocrine cells [22]. 

Our hypothesis is that the RA signaling pathway regulates HOXA9 expression and when dysregulated, promotes overpopulation of ALDH+ SC and contributes to CRC tumorigenesis. To address this hypothesis, we took a three-pronged approach: (1) the effects of ATRA and the ALDH enzyme inhibitor DEAB on HOXA9 expression and cell proliferation were measured using CRC cell lines, (2) immunohistochemical (IHC) and immunofluorescence (IF) mapping was used to determine the histologic location of HOXA9-positive cells in relation to SCs and other differentiated colonic cell types in normal and neoplastic human colonic tissues, (3) bioinformatics analyses were also performed to identify proteins that are functionally associated with HOXA9 and determine if the genes that encode these proteins contain binding sites for retinoic acid receptors (RARs) and retinoid X receptors (RXRs).

## 2. Results

### 2.1. Cell Proliferation Is Reduced by Modulating RA Signaling with ATRA and DEAB

We examined the effects of both ATRA and DEAB on the proliferation of two CRC cell lines, HT29 and SW480. Note that HT29 cells contain wild-type RAR and RXR genes, while SW480 cells have mutations in the *RARA* and *RXRG* genes [24]. First, IC50 concentrations were determined for ATRA and DEAB in HT29 and SW480 cells (Figure 1A,B). We found that the IC50 concentration for ATRA was 10 µM for both cell lines, which is consistent with prior studies [7,23,25]. The IC50 concentrations for DEAB were 100 µM for SW480 and 1 mM for HT29. Next, we performed a time course experiment up to 96 h using the IC50 values of ATRA and DEAB for HT29 and SW480 cells. The results showed that both cell lines had reduced rates of proliferation when treated with ATRA or DEAB in comparison to untreated cells (Figure 1C,D) and that the decrease in CRC cell proliferation occurred in a time-dependent manner. 

### 2.2. ATRA and DEAB Reduce the Expression of Stem Cell Markers and Increase Cell Differentiation

We conducted Nanostring mRNA profiling of HT29 cells that were treated with ATRA or DEAB. Nanostring analysis showed that ATRA treatment of HT29 cells led to the downregulation of the SC gene *ALDH1A1* and the upregulation of the enterocyte differentiation gene *KRT20* (Figure 2A). HT29 cells treated with DEAB showed downregulation of the SC marker *LGR5* and upregulation of the neuronal differentiation marker *ENO2* (Figure 2B). These changes in expression indicated that ATRA and DEAB treatments reduced the number of stem cells and promoted differentiation in HT29 cells.

### 2.3. ATRA and DEAB Reduce HOXA9 Expression

We previously reported that *HOXA9* is overexpressed in CRC [4]. To determine whether *HOXA9* expression could be regulated by RA signaling, we used bioinformatics (ALGGEN research software) to identify potential RAR and RXR transcription factor binding sites (RA regulatory elements or RAREs) in the promoter region of the *HOXA9* gene (Table 1). We found 14 RXRA transcription factor binding sites, two RARB binding sites, and one binding site for the RARB–RXRA binding complex. Furthermore, the RARB sites overlap with the RXRA binding sites.

To evaluate *HOXA9* expression in ATRA-treated HT29 cells, we used RT-qPCR analysis. The results showed that HT29 cells treated with ATRA had significantly reduced *HOXA9* expression in comparison to untreated cells, and *HOXA9* expression progressively decreased in a time-dependent manner (Figure 3A). We then assessed the level of HOXA9 protein expression by western blot and densitometry. Western blot analysis indicated that HOXA9 protein expression was reduced in cells treated with ATRA (Figure 3B,C). Western blot analysis of DEAB-treated HT29 cells also showed a decrease in HOXA9 expression in response to DEAB treatment (Figure 3D,E). In comparison to the decrease in ALDH expression found after ATRA treatment (Figure 2A), ALDH expression increased in response to DEAB treatment (Figure 3F).

### 2.4. HOXA9 Is Selectively Expressed in Colonic SCs and Overexpressed in Human Colon Cancers

To evaluate the histologic location and expression of HOXA9 protein in normal and tumor colon tissue, we conducted immunohistochemistry (IHC) staining on normal and tumor tissues from CRC patients. We found that HOXA9 had greater expression in the tumor tissue compared to normal colon (Figure 4A). Next, we performed immunofluorescence (IF) co-staining to evaluate the histologic localization and possible co-localization of HOXA9 with the SC marker ALDH, the neuroendocrine cell marker CGA, and the Paneth cell marker lysozyme (Figure 4B–E). We found not only that both HOXA9 and ALDH showed increased expression in tumor cells, but also that they were co-expressed in cells in both tissue samples. Notably, neither lysozyme nor CGA staining was observed in the neoplastic epithelium in colonic tumor tissue, but staining was observed within the stroma. This is consistent with our previous results showing that differentiated cells, particularly neuroendocrine cells, become depleted during colon tumorigenesis and that ATRA can induce differentiation of CSCs into neuroendocrine cells [22]. 

### 2.5. A Network of Proteins Associated with HOXA9 Is Classified as Being Transcriptionally Dysregulated in Cancer, and Genes Encoding Them Contain RAR and RXR Binding Sites

Our bioinformatics analysis utilizing the STRING database identified a network of 20 proteins predicted to functionally associate with HOXA9 based on their known and/or predicted interaction with HOXA9 (Figure 5A). A functional annotation chart from the DAVID analysis software showed eight genes related to RNA transport, three genes related to the TGF-β signaling pathway and the cell cycle, and another subset of three genes related to other signaling pathways regulating stem cell pluripotency (Figure 5B and Table 2). Notably, six genes—*MLLT3*, *MEIS1*, *PBX1*, *PBX3*, *HOXA10*, *KMT2A*—were found to play a role in transcriptional dysregulation in cancer.

To determine whether HOXA9-associated proteins could be regulated by RA signaling, we used bioinformatics to identify potential RAR and RXR transcription factor binding sites in the promoter region of the genes that encode these proteins. As for *HOXA9* (Table 1), we used ALGGEN to find potential RAREs within the 20 genes encoding proteins predicted to functionally interact with *HOXA9* (Table 3). We found multiple RXRA binding sites and at least one RARB binding site within the promoter region of all 20 genes. Five of these genes also appeared to contain RARB–RXRA complex binding sites, which are the same binding sites as those found within the *HOXA9* promoter.

## 3. Discussion

In our study, we investigated how *HOXA9* might contribute to the overpopulation of CSCs in CRC development. Our first result showed that both ATRA and DEAB treatments inhibited HT29 and SW480 cell proliferation. This might seem perplexing because ATRA is an activator of RA signaling, while DEAB is an inhibitor of ALDH enzymatic activity; therefore, one might expect these drugs should work in an opposite manner. However, it has been reported that both ATRA and DEAB decrease ALDH activity. For example, Croker et al. [26] noted that (i) DEAB is a competitive substrate of ALDH rather than a direct inhibitor of ALDH isozyme expression; (ii) ATRA inhibits ALDH promoter activity indirectly through the RA signaling pathway. Moreover, Croker and Allan [27] showed that treatment of human breast cancer cells (MDA-MB-231, MDA-MB-468) with either DEAB or ATRA downregulated ALDH activity. Taken together, these findings indicate that both ATRA and DEAB reduce cell proliferation by inhibiting ALDH: ATRA by decreasing ALDH1 expression, and DEAB by inhibiting ALDH enzyme activity.

We next investigated whether RA signaling regulates HOXA9 expression. Our bioinformatics analysis indicated that *HOXA9* contains RARB and RXRA transcription factor binding sites within its promoter region. Since ATRA complexes with RAR and RXR to regulate genes that have such binding sites in their promoters, this suggests that *HOXA9* is regulated by RA signaling. More specifically, ATRA likely binds to an RARB–RXRA complex to regulate *HOXA9*, and ATRA treatment should reduce *HOXA9* expression. Moreover, our IHC and IF staining showed that HOXA9 is selectively expressed in ALDH+ normal SCs and CSCs in human colonic tissues. That HOXA9 has RARE elements in its promoter region and is co-expressed with the SC marker ALDH indicates that HOXA9 is regulated by RA signaling in ALDH+ SCs. Notably, we found that treatment of CRC cells with ATRA and DEAB led to a decrease in HOXA9 expression. SC genes (*ALDH1A1* and *LGR5*) were also downregulated in HT29 cells treated with ATRA and DEAB, while differentiation genes (*KRT20* and *ENO2*) were upregulated. Our results with ATRA are consistent with previous studies showing that a decrease of *HOXA9* expression in cancer cells occurs upon exposure to ATRA [6,28]. Thus, these results support our hypothesis that *HOXA9* is regulated by the RA signaling pathway in ALDH+ stem cells. 

Next, we investigated whether dysregulated RA signaling might result in HOXA9 overexpression. Our IHC staining showed that HOXA9 is overexpressed in tumor tissue compared to normal epithelium. Our IF co-staining for HOXA9 and ALDH also revealed that: (i) HOXA9- and ALDH-co-stained cells were present within the same regions of the tumor; (ii) both HOXA9 and ALDH showed increased expression in tumor tissue compared to normal tissue. In addition to CRC, HOXA9 has been found to be overexpressed in breast cancer and leukemia [4,16]. Reduced staining for CGA+ neuroendocrine and lysozyme+ Paneth differentiated cells was found in colonic tumor tissue in comparison to normal colon. Previous studies revealed that ATRA induces differentiation of ALDH+ cells into neuroendocrine cells [21]. This suggests that overexpression of HOXA9 may inhibit the differentiation of CSCs into neuroendocrine cells (CGA) and Paneth cells (LYZ). Taken together, these results support our hypothesis that RA signaling regulates HOXA9 expression, and dysregulated RA signaling results in HOXA9 overexpression, which may contribute to the CSC overpopulation that drives CRC growth and development.

We also performed a bioinformatics analysis to identify additional genes that are associated with HOXA9 in CRC. Previously, we predicted the existence of global regulatory gene networks for HOX genes in CRC based on functional associations of aberrantly expressed HOX genes with other genes [15]. Our analysis herein identified six additional functionally associated HOXA9 proteins, (including HOXA10) which are predicted to interact with HOXA9 and are classified as having a role in transcriptional regulation in cancer. All six proteins are encoded by genes that contain potential RARB and RXRA binding sites. This could mean that all of these co-expressed genes along with *HOXA9* are regulated by the RA pathway. Indeed, MEIS1 forms complexes with HOXA9 in acute myeloid leukemia to drive carcinogenesis [29]. However, another study showed that *MEIS1* is upregulated in neuroblastoma in response to ATRA treatment [30]. Moreover, *MEIS1* inhibition is observed in BRAF-mutant colon tumors [31]. Thus, it is possible that ATRA might inhibit *MEIS1* expression via RA signaling in the colon. Other studies showed that PBX1, HOXA, and MEIS1 bind to the *MYB* gene, activating and dysregulating it in leukemia [32]. Furthermore, *PBX1* expression is reduced in neuroblastoma in response to ATRA and other retinoids [33]. *PBX3* behaves similarly, with leukemic cells experiencing reduced *PBX3* expression in response to ATRA treatments [34]. When *PBX3* and *HOXA9* are inactivated, they repress the survival of leukemic cells [35]. That all the genes encoding these proteins have retinoid receptor binding sites as found within the *HOXA9* promoter, suggests that many of the genes that encode proteins in the HOXA9 network of associated proteins might be co-regulated through RA signaling. 

While the current study provides important information on how RA signaling and the regulation of *HOXA9* expression might play a role in the SC origin of human CRC, we realize that there it has some limitations. Indeed, our Nanostring profiling on CRC cells provided a large amount of information on changes in gene expression due to treatment with ATRA and DEAB. Accordingly, in our future research, we will perform a bioinformatics analysis and follow-up validation of the results on gene expression. For example, we will perform western blot analysis of ALDH1A1, CK20, LGR5, and ENO2 as well as RA signaling components (such as RXR and RAR) and other HOX genes to confirm the results from Nanostring. We are also planning to show the impact of DEAB treatment on ALDH enzyme activity. Another limitation of this study is that treatment with ATRA can affect several downstream signaling pathways (e.g., ERK1/2). Thus, our future research will involve additional knockdown/overexpression experiments to further understand the role that HOXA9 plays in CRC development.

## 4. Materials & Methods

### 4.1. Cell Culture

HT29 and SW480 colon cancer cell lines (ATCC, Manassas, VA, USA) were grown in McCoy’s 5A and Leibovitz 15 medium (Thermofisher Scientific, Waltham, MA, USA). Both media were supplemented with 10% FBS (Atlanta Biologicals/R&D Systems, Flower Branch, GA, USA) and 1% Pen Strep (Gibco, Waltham, MA, USA) antibiotic. Cell cultures were maintained in an incubator at 37 °C with 5% CO_2_.

### 4.2. ATRA and DEAB Treatments

HT29 and SW480 cells were treated with 10 µM all-trans retinoic acid (ATRA, Stem Cell Technologies, Vancouver, BC, Canada) unless otherwise noted. DMSO at 0.1% concentration was used as a vehicle control for ATRA. HT29 and SW480 cells were treated with DEAB (Sigma, St. Louis, MO, USA) at 1 µM and 100 µM, respectively, unless otherwise noted. Ethanol at 0.1% was used as a vehicle control for DEAB. Cell medium was replaced every 48 h. All treatments were carried out in triplicate.

### 4.3. Cell Proliferation Assay

Cells were plated onto 96-well plates at 15,000 cells/well over 24, 48, 72, and 96 h. Cells were washed with 1× PBS, then stained with crystal violet dye (Sigma, St. Louis, MO, USA) for 30 min. Stained cells were then rinsed in water, and plates were left to dry overnight. Crystal violet was reconstituted in 10% acetic acid for up to one hour. The optical density of each well was measured at 570 nm using a Tecan multiplate reader and i-control software. All treatments were conducted in triplicate.

### 4.4. Western Blotting

Cells were plated onto 6-well plates at 150,000 cells/well over 24, 48, and 72 h. All treatments were performed in triplicate. Treated HT29 and SW480 cells were lysed in RIPA buffer (Sigma, St. Louis, MO, USA) plus 1× HALT protease inhibitor cocktail (Thermofisher Scientific, Waltham, MA, USA). Protein concentrations were determined using a BCA protein assay kit (Thermofisher Scientific, Waltham, MA, USA). Protein lysates containing 35 µg of protein were loaded per sample and subjecedt to SDS-PAGE using 4–20% Bio-Rad (Bio-Rad, Hercules, CA, USA) gels at 200 volts for 30 min. Protein bands were then transferred onto PVDF paper (0.45 μm, Thermofisher Scientific, Waltham, MA, USA) at 100 volts for one hour at 4°. PVDF paper was blocked in blotto solution, 5% nonfat dried milk in 1× Tris-buffered saline (Bio-Rad, Hercules, CA, USA) with 0.1% Tween (TBS-T, Thermofisher Scientific, Waltham, MA, USA) for one hour. Blots were incubated in primary antibodies diluted in 5% blotto on a rocking platform at 4° overnight. Secondary antibodies were also prepared in blotto and incubated at room temperature for one hour. Blots were washed three times with TBS-T after each antibody incubation. Protein bands were detected using Super Signal West Dura Extended Duration Substrate (Thermofisher Scientific, Waltham, MA, USA) for 5 min, followed by Imaging and densitometry using the Li-Cor Odyssey imager (LI-COR Biosciences, Lincoln, NE, USA) and Image studio software (LI-COR Biosciences, Lincoln, NE, USA). The primary antibodies used were HOXA9 (1:500, 1A12C1 Thermofisher Scientific, Waltham, MA, USA), ALDH (1:500, 611194 BD Biosciences, San Jose, CA, USA), and b-actin (1:2000, 8H10D10 Cell Signaling Technology, Danvers, MA, USA). The secondary antibody used was m-IgGk binding protein conjugated to horseradish peroxidase (1:1000, sc-516102 Santa Cruz Biotechnology, Dallas, TX, USA).

### 4.5. Nanostring Profiling

Cells were plated onto 6-well plates at 150,000 cells/well. Cells were treated with either ATRA or DEAB and their respective controls, DMSO or EtOH, for 96 h. All treatments were performed in triplicate. Next, cells were lifted off the plate with trypsin and resuspended in media. Cells were then spun down and resuspended in Trizol LS reagent (Thermofisher Scientific, Waltham, MA, USA). Once in Trizol, the cells were vortexed for 30 s, incubated at room temperature for 10 min, and stored at −80 °C. Once an *n* = 3 was obtained, the frozen samples were sent to the Wistar Institute Genomics Core Facility for Nanostring profiling (Nanostring Technologies, Seattle, WA, USA). The “pan-cancer Nanostring panel” that contains 800 targets was chosen for our profiling. The Nanostring data was analyzed using nSolver software (Nanostring Technologies, Seattle, WA, USA).

### 4.6. Immunohistochemistry (IHC) and Immunofluorescence (IF) Microscopy

We obtained 5 µm formalin-fixed, paraffin-embedded sections of human colonic tissues from Christiana Care Health System. Tissue samples involved matched normal and malignant colonic tissue pairs (*n* = 5 or more tissue pairs). Specimens were deparaffinized using Citri Solv (Thermofisher Scientific, Waltham, MA, USA) solution and rehydrated in ethanol. Next, specimens were quenched for peroxidase activity in 3% H_2_O_2_ in methanol. Antigen retrieval was performed in 1× Antigen Retrieval Solution (pH 6.0, Thermofisher Scientific, Waltham, MA, USA) at 95 °C for approximately 30 min. Slides were allowed to cool to room temperature and washed in distilled H_2_O. Next, specimens were soaked in a blocking solution for 10 min and incubated in primary antibody overnight at 4 °C. Slides were then washed 3× in TBS-T and incubated in secondary antibodies at room temperature for 2 hours followed by an additional 3 washes in TBS-T. IHC slides were then stained with metal-enhanced DAB (Thermofisher Scientific, Waltham, MA, USA) and counterstained with hematoxylin. Stained IHC slides were then dehydrated in 90–100% ethanol and in Citra solv solution before applying mountant. IF slides were instead stained and mounted with ProLong Glass Antifade Mountant with NucBlue Stain (Thermofisher Scientific, Waltham, MA, USA). IF Images were taken with a ZEISS Epi-fluorescence microscope (Zeiss Microscopy, Jena, Germany) and Zen software (Zeiss Microscopy, Jena, Germany). The primary antibodies used were HOXA9 (1:100, ab191178 Abcam, Cambridge, UK), CGA (1:100, M0869 Dako, Copenhagen, Denmark), ALDH (1:400, PA5-18483 Invitrogen, Carlsbad, CA, USA), and Lysozyme (1:400, MA5-36114 Thermofisher Scientific, Waltham, MA, USA). The IgG primary antibody controls used were rabbit-IgG (1:250, 37415 Abcam, Cambridge, UK), goat-IgG (1:1000, 37373 Abcam, Cambridge, UK), and mouse-IgG (1:100 14-4714-82 Thermofisher Scientific, Waltham, MA, USA). For the IHC stain, we used an anti-rabbit secondary antibody (1:200, sc-2357 Santa Cruz Biotechnology, Dallas, TX, USA). For the IF stains, we used ALEXAFLUOR secondary antibodies 488 anti-rabbit (1:100, A11008 Invitrogen, Carlsbad, CA, USA), 594 anti-mouse (1:100, A21203 Invitrogen, Carlsbad, CA, USA), and 647 anti-goat (1:100, A21447 Invitrogen, Carlsbad, CA, USA).

### 4.7. Bioinformatics

ALGGEN Research Software [36,37] was used to identify transcription factor binding sites. Proteins predicted to interact with HOXA9 were identified using the STRING: functional protein association networks (https://string-db.org, accessed on 2 November 2021) and database [38]. The input for the analysis was simply “HOXA9”. Details on the bioinformatics results with the scores and any additional information are available upon request by contacting the corresponding author. The identified proteins were categorized based on a functional annotation chart made with the DAVID bioinformatics tool [39,40]. The promoters of the genes were found using the NCBI database (https://www.ncbi.nlm.nih.gov/, accessed on 2 November 2021).

### 4.8. RNA Isolation, cDNA Conversion, and qPCR

Cells were grown in 6-well plates at 150,000 cells/well over 24, 48, and 72 h. Six replicates were used in the experiment. Treated HT29 cells were lysed in Trizol LS reagent (Thermofisher Scientific, Waltham, MA, USA). Chloroform was then used to isolate the mRNA, which was then washed with isopropanol followed by ethanol before the mRNA is resuspended in RNAse-free water. Excess DNA was removed using a DNA removal kit (Thermofisher Scientific, Waltham, MA, USA). First, DNase was added to the RNA samples, which were incubated at 37 °C. The DNase inactivation reagent was then added to the samples to stop the process.

The mRNA was then converted to cDNA via a cDNA conversion kit (Invitrogen, Carlsbad, CA, USA). Samples were incubated at 65 °C with oligo primers (Invitrogen, Carlsbad, CA, USA) and a dNTP mix (Thermofisher Scientific, Waltham, MA, USA) to anneal the primers. After cooling, RNase inhibitor (Thermofisher Scientific, Waltham, MA, USA), 0.1M DTT, and SuperScript III reverse transcriptase (Thermofisher Scientific, Waltham, MA, USA) were added to the sample, which was heated to 50 °C to produce the cDNA, then heated at 70 °C to stop the process. Controls without Superscript III reverse transcriptase were also used to check for contamination.

SYBR green PCR Master Mix (Thermofisher Scientific, Waltham, MA, USA) and respective primer pairs were added to samples. Samples were then placed into a 7500 Fast Real Time PCR System (Thermofisher Scientific, Waltham, MA, USA). PCR was run for 40 cycles, t 95 °C for denaturation and 60 °C for annealing and extension. The primers used targeted beta II tubulin (HP203790, Origene, Rockville, Maryland) and HOXA9 (HP226545, Origene, Rockville, Maryland).

### 4.9. Statistical Analysis

*t*-tests were performed to determine the significance of RT-qPCR and western blot results. Statistical analysis for Nanostring profiling was performed using Nsolver software (Nanostring Technologies, Seattle, WA, USA).

## 5. Conclusions

The goal of this study was to ascertain how RA signaling and regulation of *HOXA9* expression might play a role in the SC origin of human CRC. Previous research in our lab reported that HOXA9 is selectively expressed in ALDH+ CSCs and overpopulation of CSCs drives CRC development and growth. Moreover, we found that the retinoid receptors RXR and RAR are selectively expressed in ALDH+ SCs, which indicated RA signaling mainly occurs via ALDH+ SCs [22,41]. Thus, we surmised that RA signaling might regulate HOXA9 gene expression in colonic ALDH+ SCs. Our results herein support our hypothesis that RA signaling regulates HOXA9 expression, and dysregulated RA signaling results in HOXA9 overexpression, which contributes to CSC overpopulation in CRC. These findings provide a key mechanism for the design of novel anti-CSC-targeted treatments for CRC. For example, the inhibition of ALDH1A sensitizes CRC cells to the anti-tumor effects of chemotherapy [42]. Overall, our study showing that RA signaling of HOX genes plays a role in the regulation of stem cells provides a regulatory mechanism that might be useful to develop treatments against CSC overpopulation in CRC.

## Figures and Tables

**Figure 1 ijms-23-06799-f001:**
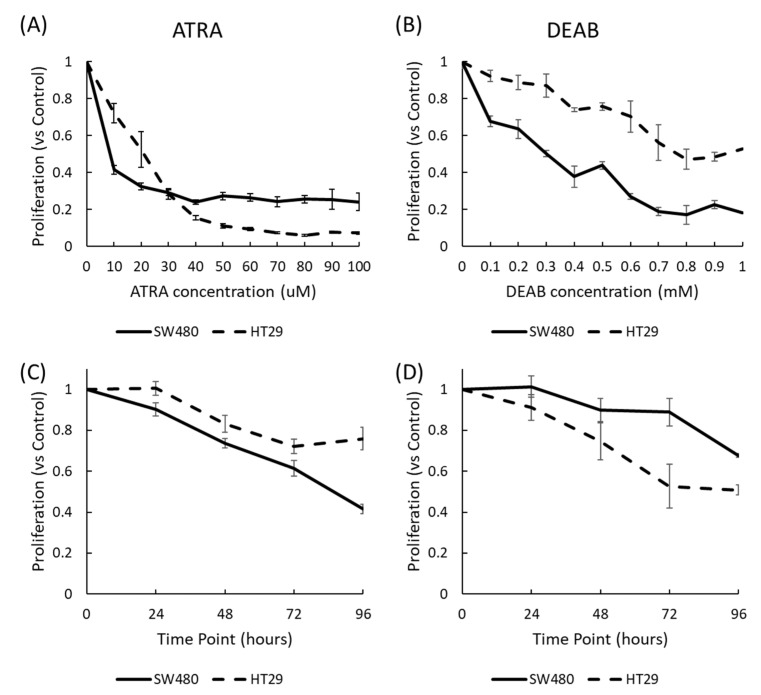
Modulation of RA signaling with ATRA and DEAB decreases CRC cell proliferation. ATRA- (**A**) and DEAB- (**B**) treated cells at varying concentrations at 96 h to determine IC50 values. HT29 and SW480 cells treated for longer times showed reduced rates of proliferation (**C**,**D**). Vehicle controls were DMSO for ATRA-treated cells and EtOH for DEAB-treated cells.

**Figure 2 ijms-23-06799-f002:**
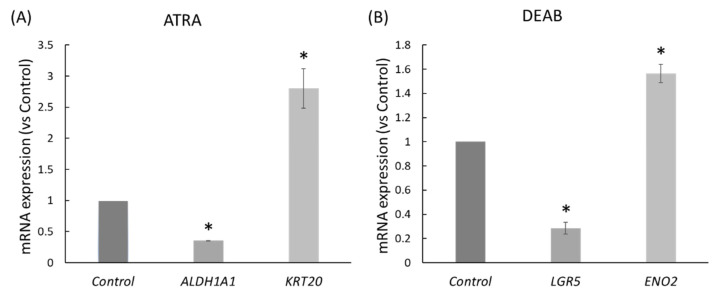
ATRA and DEAB treatments reduce the expression of stem cell markers and increase the expression of markers of cell differentiation in HT29 cells. Nanostring analysis of significant (*p* < 0.05) changes in the expression of ALDH1A1, KRT20, LGR5, and ENO2 in CRC cells treated with ATRA (**A**) and DEAB (**B**) at IC50 concentration for 96 h. Asterisks indicate significance compared to vehicle control treatment (*p* ≤ 0.05).

**Figure 3 ijms-23-06799-f003:**
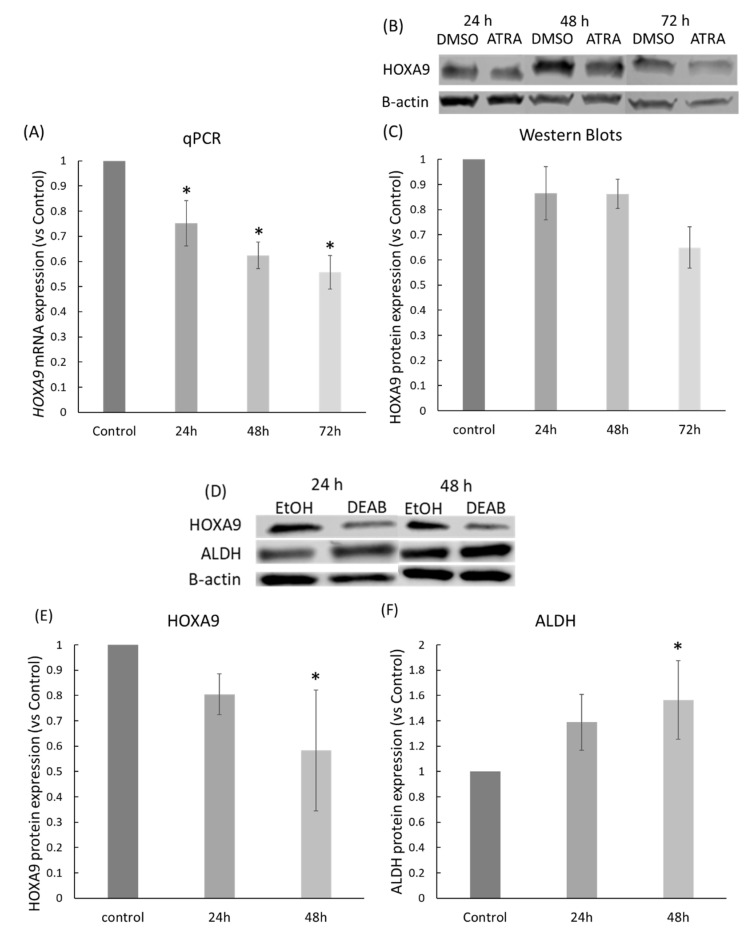
ATRA and DEAB treatments of HT29 CRC cells decrease HOXA9 expression. (**A**) RT-qPCR of HT29 cells treated with ATRA at IC50 concentration for 24, 48, and 72 h. Expression of *HOXA9* according to RT-qPCR results was normalized to that of B-tubulin; (**B**) western blot of HT29 cells treated with ATRA at IC50 concentration for 24, 48, and 72 h. Densitometry of western blot (**C**) represents adjusted mean ratio of reduced HOXA9 expression in ATRA-treated cells; (**D**) western blot of ATRA- and DEAB-treated HT29 cells at IC50 concentrations for 24 and 48 h. Densitometry of the blots (**E**,**F**) represents the adjusted mean ratio of changes in HOXA9 and ALDH expression in ATRA- and DEAB-treated cells. All western blots were normalized to β-actin and vehicle control expression. Asterisks indicate significance compared to vehicle control (*p* ≤ 0.05).

**Figure 4 ijms-23-06799-f004:**
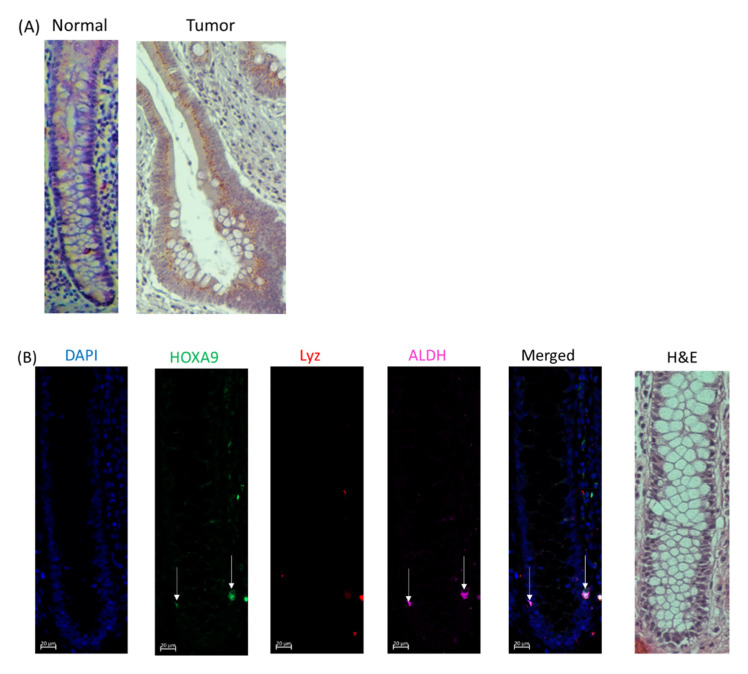
HOXA9 protein is selectively expressed in ALDH+ colonic SCs and is overexpressed in CRCs. (**A**) Images of normal and tumor colon tissue stained for HOXA9. Normal tissue is at 10× magnification, and tumor tissue is at 20× magnification. The tumor shown is a stage 2A adenocarcinoma of the ascending colon. IF staining was performed on normal and tumor tissues (**B**,**C**) that were also co-stained for HOXA9 (green), lysozyme (red), and ALDH (magenta). The tumor shown is a stage 2A mucinous adenocarcinoma of the descending colon. IF images are also shown of normal and tumor colon tissues (**D**,**E**) that were co-stained for HOXA9 (green), CGA (red), and ALDH (magenta). The tumor shown is a stage 1 mucinous adenocarcinoma in the ascending colon. Arrows point to ALDH+ cells. All IF images are at 20× magnification. Separate hematoxylin and eosin (H&E) slides are included to show the general tissue structure in matching IF images.

**Figure 5 ijms-23-06799-f005:**
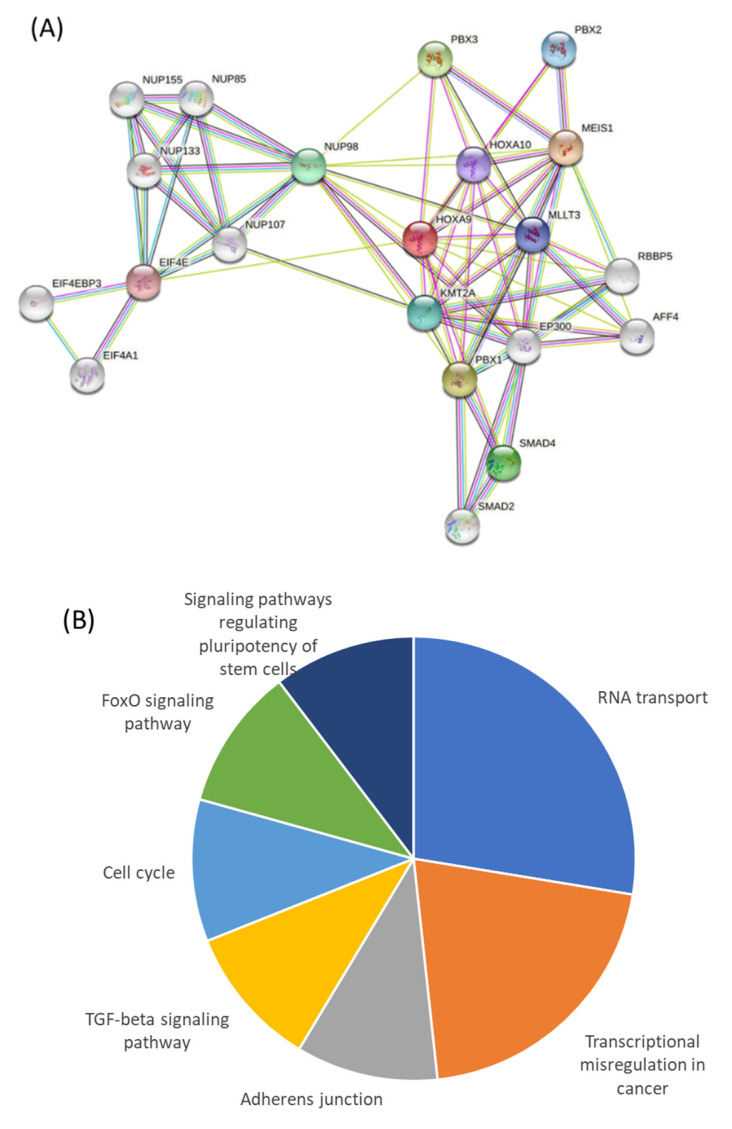
Proteins predicted to interact with HOXA9 and their functional classification. (**A**) The image shows the 20 proteins that were identified to have known or strongly predicted interactions with the HOXA9 protein. These proteins were organized into groups based on known functions ((**B**); Table 2), using the David analysis software.

**Table 1 ijms-23-06799-t001:** List of RAR and RXR binding sites found in the HOXA9 promoter.

Transcription Factor	Start Position *	End Position	TRE ** Sequence
RXR-alpha	−1454	−1448	TTTACCC
RXR-alpha	−1442	−1436	GGGTGAG
RXR-alpha	−1380	−1374	TTTACCC
RXR-alpha	−1168	−1162	GGGTCTC
RXR-alpha	−1021	−1015	GGGTCGC
RXR-alpha	−945	−939	GGGTCGA
RXR-alpha	−843	−837	GGGTCCT
RXR-alpha	−816	−810	GGGTTGC
RXR-alpha	−787	−781	GGGTAGC
RXR-alpha	−770	−764	TACACCC
RXR-alpha	−654	−648	GGGTAAA
RXR-alpha	−346	−340	GGGACCC
RXR-alpha	−205	−199	CAAACCC
PPAR-alpha:RXR-alpha	−1413	−1403	CAGGCCCAGGA
PPAR-alpha:RXR-alpha	−1025	−1015	GGCTGGGTCGC
RAR-beta	−817	−808	AGGGTTGCCC
RAR-beta	−207	−198	AACAAACCCC
RAR-beta:RXR-alpha	−42	−31	GGGCGCCGGCAA

* The first base pair of the transcribed gene is position 0. ** TRE = transcriptional regulatory element.

**Table 2 ijms-23-06799-t002:** List of genes under each functional classification as described in Figure 5B.

Functional Classification	Genes
RNA transport	*EIF4A1, EIF4EBP3, EIF4E, NUP107, NUP133, NUP155, NUP85, NUP98*
Transcriptional misregulation in cancer	*MLLT3, MEIS1, PBX1, PBX3, HOXA10, KMT2A*
Adherens junction	*EP300, SMAD2, SMAD4*
TGF-beta signaling pathway	*EP300, SMAD2, SMAD4*
Cell cycle	*EP300, SMAD2, SMAD4*
FoxO signaling pathway	*EP300, SMAD2, SMAD4*
Signaling pathways regulating pluripotency of stem cells	*MEIS1, SMAD2, SMAD4*

**Table 3 ijms-23-06799-t003:** List of RAR and RXR binding sites in the promoter of genes classified as being transcriptionally dysregulated in cancer and functionally associated with HOXA9.

Transcription Factor	*MLLT3*	*MEIS1*	*HOXA10*	*PBX1*	*PBX3*	*KMT2A*	*EIF4E*	*NUP98*	*SMAD4*	*PBX2*	*NUP115*	*NUP85*	*NUP133*	*NUP107*	*EIF4EBP3*	*EIF4A1*	*RBBP5*	*AFF4*	*EP300*	*SMAD2*
RXRA	11	15	21	4	8	13	8	15	7	18	11	11	8	5	10	12	9	14	15	13
RARB	2	4	7	1	1	6	1	4	2	4	4	5	4	2	4	3	1	4	4	3
RARB: RXRA	1	2	4		3	1	1	2	1	4	1			2	3		1	2		2
PXR-1/ RXRA	3		2	3		2	3	2	1	3	1	3	2	3	4	1	3	3	1	3
PPARA/ RXRA		2	1		1	3	1	4	4	2	1	2	1	1	4		2	3		2
RARA1		1						2												
**TOTAL**	17	24	35	8	13	25	14	29	15	31	18	21	15	13	25	16	16	26	20	23

## Data Availability

The link to the publicly archived datasets analyzed or generated during the study is available from STRING functional protein association networks (https://string-db.org, accessed on 2 November 2021) and the NCBI database (https://www.ncbi.nlm.nih.gov/, accessed on 2 November 2021).

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
