# Peer review of "HOXA9 Overexpression Contributes to Stem Cell Overpopulation That Drives Development and Growth of Colorectal Cancer"

_ijms, 2022, doi:10.3390/ijms23126799_

Round 1

Reviewer 1 Report

Thank you for providing an opportunity to review the manuscript titled “HOXA9 overexpression contributes to stem cell overpopulation that drives development and growth of colorectal cancer” by Brian Osmond, et. al.

In this report, Brian Osmond and co-authors investigates the role of HOXA9 overexression on colorectal cancer development and also try to explain the mechanistic role of RA pathway in the regulation of HOXA9 in the CRC growth and development. They have also conducted bioinformatic analysis to identify the RA receptor binding sites in HOXA9 and its binding partners, which might provide a clue on RA pathway in the development of CRC. Although the manuscript provides valuable information on the effect of RA modulation on HOXA9, a more rigorous experimental approach is needed to validates this point. Please find the comments below:

Major comments:

  1. If the ATRA is an activator of RA signaling, and the ALDH inhibitor DEAB is an inhibitor to RA signaling, one would expect these drugs works in an opposite manner unless strong experimental proofs provided.
  2. Since the cell line HT29 have wt RAR/RXR genes and the SW480 have the mutated RAR/RXR genes, how would you explain the similar proliferation profile observed when these cells were treated with ATRA or DEAB. Does this means these drugs works outside the RA signaling pathway to modulate the proliferation?
  3. Does the ATRA exerts its effect on HOXA9 transcriptionally? Since the authors have shown that the HOXA9 promoter have several RAR-RXR binding sites, which may get occupied after the treatment of ATAR (which activates RA signaling), it must be essential to know if this leads to a suppression of HOXA9 transcription. Does the Nanostring profiling measured the levels of HOXA9 transcript?Alternatively, a reporter assay to prove the effect of ATRA or DEAB on HOXA9 promoter activity can be conducted.
  4. The suppressive effect of ATRA on HOXA9 protein expression is not seems statistically significant (Fig.2C). If yes, please provide the p-values. Have you tested the effect of inhibition of ALDH using DEAB on the protein levels of HOXA9? It may be essential to show, since the IF staining shows higher expression of HOXA9 in the ALDH+ CSC tissue.

Minor comments:

  1. Labeling of cell line names not consistent in the Fig1. For eg. Fig 1d uses small letters to label cell lines as sw480 and ht29, whereas, Fig1a,b and c uses capital letters SW480 or HT29.

Author Response

Response to reviewer1 comments

We thank the expert reviewers for their review of our manuscript. We have made major revisions as requested and revised our manuscript according to the reviewers' helpful comments. Our responses (â–º) to each comment are given below

REVIEWER 1.

Comment 1. If the ATRA is an activator of RA signaling, and the ALDH inhibitor DEAB is an inhibitor to RA signaling, one would expect these drugs works in an opposite manner unless strong experimental proofs provided.

Response â–ºThank you for this important question. Our findings indicate that both ATRA and DEAB reduce cell proliferation by inhibiting ALDH; ATRA by decreasing ALDH1 expression and DEAB by inhibiting ALDH enzyme activity. We have added a paragraph to the discussion to answer this question.

Comment 2. Since the cell line HT29 have wt RAR/RXR genes and the SW480 have the mutated RAR/RXR genes, how would you explain the similar proliferation profile observed when these cells were treated with ATRA or DEAB. Does this means these drugs works outside the RA signaling pathway to modulate the proliferation?

Response â–ºWe have found that proliferation of both HT29 cells and SW480 cells is inhibited by ATRA, but the SW480 cell are somewhat resistant so that they require higher doses of ATRA to inhibit proliferation. The resistance to ATRA is attributed to the mutations in RARA & RXRG in SW480 cells.

Comment 3. Does the ATRA exerts its effect on HOXA9 transcriptionally? Since the authors have shown that the HOXA9 promoter have several RAR-RXR binding sites, which may get occupied after the treatment of ATAR (which activates RA signaling), it must be essential to know if this leads to a suppression of HOXA9 transcription. Does the Nanostring profiling measured the levels of HOXA9 transcript? Alternatively, a reporter assay to prove the effect of ATRA or DEAB on HOXA9 promoter activity can be conducted.

â–ºOur findings indicate that the ATRA effect is on HOXA9 transcription. We have now measured the response of ATRA on HT29 and SW480 CRC cells at both the protein level and mRNA level. A reporter assay is a good idea and we have incorporated that into our next study.

Comment 4. The suppressive effect of ATRA on HOXA9 protein expression is not seems statistically significant (Fig.2C). If yes, please provide the p-values. Have you tested the effect of inhibition of ALDH using DEAB on the protein levels of HOXA9? It may be essential to show, since the IF staining shows higher expression of HOXA9 in the ALDH+ CSC tissue.

â–ºWe have now done additional experiments to test the effect of DEAB on protein levels of HOXA9 and have added the statistical significance to the figures.

Comment 5. Labeling of cell line names not consistent in the Fig1. For eg. Fig 1d uses small letters to label cell lines as sw480 and ht29, whereas, Fig1a,b and c uses capital letters SW480 or HT29.

â–ºWe have made the labeling of cell lines in the Figures consistent.

Reviewer 2 Report

In this article, Osmond and colleagues aim to unravel the association between RA signaling and HOXA9 in colorectal cancer. Overall, this is an interesting study but there are several issues that I would like to bring into the attention of the authors in order to improve the current version of their work.

”Materials and Methods” require better description or more details in some cases. First of all, in this study the authors are using colonic tissues from patients but no details are provided with regards to the clinical characteristics (at least what type of samples, the stage and grade), the number of the tissue samples analyzed, the percentage of the cancer cells and the controls they used (e.g. adjacent normal tissue). Also, there is no statement with regards to the Ethics. Approval is required prior publication of the study and the ethics approval and patients’ consent should be clearly described in your manuscript. Please provide all this information. Also, in Material and Methods, statistics is missing, maybe can be combined with bioinformatics section? A clear description of the statistical methods applied in each experiment is required. Furthermore, more details are required for the Nanostring profiling. Which are the 800 targets that were selected or which gene panel was selected and based on what criteria?

Can the authors better elaborate on the selection of HOXA9 for further study? For example, in their previous publication (PMID: 30154863) the authors found that HOXB9 was the most upregulated gene at all stages in CRC. Why did they select HOXA9 instead of HOXB9?

Furthermore, the association of ATRA and HOXA9 with proliferation has been previously described by the authors (PMID: 30410666) in the same colon cancer cell lines studied here so this is not novel. The authors should emphasize on the novelty of this study and better describe their novel findings.

What is the status of RA signaling in colorectal cancer? Is it overactivated? Experimental evidence is required. RA signaling should be checked in both colon cancer cell lines but also in primary colonic epithelial cells or benign colonic cell lines. How was it confirmed that the RA signaling was successfully affected after ATRA and DEAB treatment? Also, why the IC50 for DEAB is 1mM for HT29?

Why Nanostring was selected instead of RNA seq? Was HOXA9 included in the differentially expressed (DE) genes identified through Nanostring profiling? Which other features having the same motifs were detected as differentially expressed? The list of the deregulated mRNAs upon treatment should be provided at least as Supplementary Table and a better description of the Nanostring findings is required. Can the authors perform network and pathway analysis for these genes? Does such an analysis confirm a deregulation of RA signaling and what other pathways are found deregulated upon treatment? Such analyses are strongly recommended.

Treatment with ATRA does not have an impact solely in HOXA9 so the observed decrease in the proliferation can be a result of other proteins as well. Furthermore, based on the WB it appears that decreased HOXA9 levels are detected at 72h and not earlier. How can the authors justify the observed impact on proliferation before the 72 hours? ATRA has been already associated with suppression of the proliferation rate colon cancer cells via ERK1/2. Did the authors detect any changes in this signaling pathway? Thus, treatment with ATRA affects several signaling pathways downstream and any conclusion about HOXA9 implication requires additional knockdown/overexpression experiments. Immunoprecipitation experiments are further needed but not required.

For HOXA9 do the authors have the western blot from the 96 hours since this was the time point used for Nanostring profiling? Also, along with HOXA9 Western blot analysis for ALDH1A1, KRT20, LGR5, and ENO2 can be also included to confirm the results from Nanostring. Also, what about the expression levels of the RXR and RAR?

For the STRING analysis it is not clear what was used as input. Also, about the STRING analysis the authors describe functionally associated proteins but this is not correct. These are only PREDICTED FUNCTIONAL PARTNERS. A supporting table with the scores and additional information (coexpression, experiments, databases, textmining) can be exported from STRING and provided as Supplementary table.

In the Conclusions, I don’t agree with the following sentence: “The goal of this study was to determine how HOXA9-regulatory mechanisms might play a role in the SC origin of CRC.” RA signaling and the axis through HOXA9 association with colon cancer was the main goal of this study. The HOXA9 mediated mechanisms have not been studied. Also the experiments were not performed in FACS sorted ALDH+ CSC cells that would support this statement. Rephrasing in many parts of the manuscript is strongly recommended.

Some vague expression are used and it is strongly recommended the authors to tone down e.g “This suggests that overexpression of HOXA9 inhibits differentiation of CSCs into neuroendocrine cells (CGA) and Paneth cells (LYZ).”; “That the genes encoding these proteins all have retinoid receptor binding sites as found within the HOXA9 promoter, suggests that many if not most genes that encode proteins in the HOXA9 network of associated proteins are co-regulated through RA signaling.”; “Our study provides an important mechanism”. The authors mention in the results “that HOXA9 overexpression contributes to overpopulation of ALDH+ CSCs.” This is a vague statement that requires further experimental investigation.

How can the findings from this study be applicable for therapy? Inhibition of RA signaling can have therapeutic potential in CRC? It would be nice if you could write 1-2 sentences in the discussion.

Limitations of the study are required.

Author Response

REVIEWER 2.

Comment 1. ”Materials and Methods” require better description or more details in some cases. First of all, in this study the authors are using colonic tissues from patients but no details are provided with regards to the clinical characteristics (at least what type of samples, the stage and grade), the number of the tissue samples analyzed, the percentage of the cancer cells and the controls they used (e.g. adjacent normal tissue). Also, there is no statement with regards to the Ethics. Approval is required prior publication of the study and the ethics approval and patients’ consent should be clearly described in your manuscript. Please provide all this information. Also, in Material and Methods, statistics is missing, maybe can be combined with bioinformatics section? A clear description of the statistical methods applied in each experiment is required. Furthermore, more details are required for the Nanostring profiling. Which are the 800 targets that were selected or which gene panel was selected and based on what criteria?

â–ºWe have now added details on the clinical samples, controls, IRB approval, statistics, and Nanostring profiling.

Comment 2. Can the authors better elaborate on the selection of HOXA9 for further study? For example, in their previous publication (PMID: 30154863) the authors found that HOXB9 was the most upregulated gene at all stages in CRC. Why did they select HOXA9 instead of HOXB9?

â–ºWe selected HOXA9 because our previous study of transcriptional regulation of HOX genes (J Cell Physiol. 2019;234:13042–13056) indicated that the expression of HOXA9 and several co-expressed ALDH genes is regulated by the RA signaling pathway.

Comment 3. Furthermore, the association of ATRA and HOXA9 with proliferation has been previously described by the authors (PMID: 30410666) in the same colon cancer cell lines studied here so this is not novel. The authors should emphasize on the novelty of this study and better describe their novel findings.

â–ºWe have revised the paper to emphasize the novelty of the findings and to reference our previous work.

Comment 4. What is the status of RA signaling in colorectal cancer? Is it overactivated? Experimental evidence is required. RA signaling should be checked in both colon cancer cell lines but also in primary colonic epithelial cells or benign colonic cell lines. How was it confirmed that the RA signaling was successfully affected after ATRA and DEAB treatment? Also, why the IC50 for DEAB is 1mM for HT29?

â–ºWe have recently published that RA and WNT signaling are linked in CRC cells and APC mutation leads to activation of WNT signaling and decreased RA signaling (PLoS ONE 15(10): e0239601).

Comment 5. Why Nanostring was selected instead of RNA seq? Was HOXA9 included in the differentially expressed (DE) genes identified through Nanostring profiling? Which other features having the same motifs were detected as differentially expressed? The list of the deregulated mRNAs upon treatment should be provided at least as Supplementary Table and a better description of the Nanostring findings is required. Can the authors perform network and pathway analysis for these genes? Does such an analysis confirm a deregulation of RA signaling and what other pathways are found deregulated upon treatment? Such analyses are strongly recommended.

â–ºWe selected Nanostring profiling because in our experience it provides more accurate results on gene expression than profiling using arrays. We have found that our Nanostring profiling on CRC cells has provided a huge amount of information. Accordingly, in a separate study, we are doing extensive bioinformatics analysis and follow up validation of the results on gene expression. We anticipate that work will take several years and is beyond the scope of the current study. This future analysis is being done on the entire set of HOX genes and it is premature to provide details on the individual HOX genes at this time.

Comment 6. Treatment with ATRA does not have an impact solely in HOXA9 so the observed decrease in the proliferation can be a result of other proteins as well. Furthermore, based on the WB it appears that decreased HOXA9 levels are detected at 72h and not earlier. How can the authors justify the observed impact on proliferation before the 72 hours? ATRA has been already associated with suppression of the proliferation rate colon cancer cells via ERK1/2. Did the authors detect any changes in this signaling pathway? Thus, treatment with ATRA affects several signaling pathways downstream and any conclusion about HOXA9 implication requires additional knockdown/overexpression experiments. Immunoprecipitation experiments are further needed but not required.

â–ºWe agree that the effect from ATRA is not solely do to an impact on ATRA. This is one reason why we did bioinformatics analysis to identify a set of HOXA9 interacting proteins and to see if the genes that encode this network might be regulated by RA signaling. Clearly, additional work involving additional knockdown/overexpression and immunoprecipitation experiments and reporter assays is needed – much of this work is ongoing in our laboratory by several graduate students and post docs. We have now done RT-qPCR analysis to show that HOXA9 expression levels are decreased at 24 hours which occurs earlier than the decrease in HOXA9 protein. We have not looked at ERK1/2 yet.

Comment 7. For HOXA9 do the authors have the western blot from the 96 hours since this was the time point used for Nanostring profiling? Also, along with HOXA9 Western blot analysis for ALDH1A1, KRT20, LGR5, and ENO2 can be also included to confirm the results from Nanostring. Also, what about the expression levels of the RXR and RAR?

â–ºWe selected the 96 hour time point based on our time course for treatment effects on cell proliferation. We did not include this time point in our western analysis. We are analyzing RXR and RAR expression levels in a separate study.

Comment 8. For the STRING analysis it is not clear what was used as input. Also, about the STRING analysis the authors describe functionally associated proteins but this is not correct. These are only PREDICTED FUNCTIONAL PARTNERS. A supporting table with the scores and additional information (coexpression, experiments, databases, text mining) can be exported from STRING and provided as Supplementary table.

â–ºThe input in our STRING analysis was “HOXA9”. We apologize regarding the wording and have corrected it to state that it is “predicted”. We agree this information could be valuable to some individuals but rather than putting it in a supplemental section we decided to make it available by stating in the text that “Details on the bioinformatics results with the scores and any additional information is available upon request by contacting the corresponding author”.

Comment 9. In the Conclusions, I don’t agree with the following sentence: “The goal of this study was to determine how HOXA9-regulatory mechanisms might play a role in the SC origin of CRC.” RA signaling and the axis through HOXA9 association with colon cancer was the main goal of this study. The HOXA9 mediated mechanisms have not been studied. Also the experiments were not performed in FACS sorted ALDH+ CSC cells that would support this statement. Rephrasing in many parts of the manuscript is strongly recommended.

â–ºThank you. The wording for goal of our study has been changed as recommended.

Comment 10. Some vague expression are used and it is strongly recommended the authors to tone down e.g “This suggests that overexpression of HOXA9 inhibits differentiation of CSCs into neuroendocrine cells (CGA) and Paneth cells (LYZ).”; “That the genes encoding these proteins all have retinoid receptor binding sites as found within the HOXA9 promoter, suggests that many if not most genes that encode proteins in the HOXA9 network of associated proteins are co-regulated through RA signaling.”; “Our study provides an important mechanism”. The authors mention in the results “that HOXA9 overexpression contributes to overpopulation of ALDH+ CSCs.” This is a vague statement that requires further experimental investigation.

â–ºWe have worked to tone down the wording of the text in our revised manuscript

Comment 11. How can the findings from this study be applicable for therapy? Inhibition of RA signaling can have therapeutic potential in CRC? It would be nice if you could write 1-2 sentences in the discussion.

â–ºAgree. We have added a couple of sentences on clinical implications to the conclusion.

Round 2

Reviewer 1 Report

The authors have addressed the concerns adequately and will recommend the manuscript for publication in its current form.

Author Response

We thank Reviewer 1 for helpful comments that contributed to the quality of our manuscript.

Reviewer 2 Report

The explanation that the authors provide in the first 2 paragraphs of the discussion about the same observed impact of ATRA and DEAB in the cells is complicated and confusing. Based on the literature it seems that both ATRA and DEAB decrease ALDH activity. For example, in PMID: 28937653 it is stated “Rather than being direct inhibitors of ALDH isozyme expression, DEAB is a competitive substrate of ALDH and ATRA inhibits ALDH promoter activity indirectly through the retinoic acid pathway” and also in PMID: 21818590 “treatment of unsorted MDA-MB-231 or MDA-MB-468 cells with either DEAB or ATRA resulted in significant downregulation of ALDH activity for 24 and 48 h”. As such, I recommend the authors to improve on the explanation of the observed impact of ATRA or DEAB treatment. It would be nice to show the impact of the treatment in ALDH activity.

The limitation section is still missing. Please include in this section all of my comments that were not addressed but are planned to be checked (Comment 5, 6 and 7).

Author Response

We thank the expert reviewers for their review of our manuscript. We have made revisions as requested and revised our manuscript according to the reviewers' helpful comments. Our responses (â–º) to each comment are given below

Reviewer 2.

Comment 2.1. The explanation that the authors provide in the first 2 paragraphs of the discussion about the same observed impact of ATRA and DEAB in the cells is complicated and confusing. Based on the literature it seems that both ATRA and DEAB decrease ALDH activity. For example, in PMID: 28937653 it is stated “Rather than being direct inhibitors of ALDH isozyme expression, DEAB is a competitive substrate of ALDH and ATRA inhibits ALDH promoter activity indirectly through the retinoic acid pathway” and also in PMID: 21818590 “treatment of unsorted MDA-MB-231 or MDA-MB-468 cells with either DEAB or ATRA resulted in significant downregulation of ALDH activity for 24 and 48 h”. As such, I recommend the authors to improve on the explanation of the observed impact of ATRA or DEAB treatment. It would be nice to show the impact of the treatment in ALDH activity.

Response 2.1 â–º Thank you. We agree that our discussion about the same observed impact of ATRA and DEAB in the cells was complicated and confusing. The first paragraph has been revised to improve on the explanation of our observed impact of ATRA or DEAB treatment including citation of the two papers noted by the reviewer.

Comment 2.2. The limitation section is still missing. Please include in this section all of my comments that were not addressed but are planned to be checked (Comment 5, 6 and 7).

Response 2.2 â–º Thank you for noting our omission of study limitations. A discussion of the limitations of our study is now added to the last paragraph of the discussion section including the reviewers comments that were not addressed, but were planned to be checked in comments 5, 6, 7.